# Carnosine as a Possible Drug for Zinc-Induced Neurotoxicity and Vascular Dementia

**DOI:** 10.3390/ijms21072570

**Published:** 2020-04-07

**Authors:** Masahiro Kawahara, Yutaka Sadakane, Keiko Mizuno, Midori Kato-Negishi, Ken-ichiro Tanaka

**Affiliations:** 1Department of Bio-Analytical Chemistry, Faculty of Pharmacy, Musashino University, Tokyo 202-8585, Japan; mnegishi@musashino-u.ac.jp (M.K.-N.); k-tana@musashino-u.ac.jp (K.T.); 2Graduate School of Pharmaceutical Sciences, Suzuka University of Medical Science, Suzuka 513-8670, Japan; sadapon@suzuka-u.ac.jp; 3Department of Forensic Medicine, Faculty of Medicine, Yamagata University, Yamagata 990-9585, Japan; konoha_hamham@yahoo.co.jp

**Keywords:** zinc, ischemia, neurotoxicity, calcium

## Abstract

Increasing evidence suggests that the metal homeostasis is involved in the pathogenesis of various neurodegenerative diseases including senile type of dementia such as Alzheimer’s disease, dementia with Lewy bodies, and vascular dementia. In particular, synaptic Zn^2+^ is known to play critical roles in the pathogenesis of vascular dementia. In this article, we review the molecular pathways of Zn^2+^-induced neurotoxicity based on our and numerous other findings, and demonstrated the implications of the energy production pathway, the disruption of calcium homeostasis, the production of reactive oxygen species (ROS), the endoplasmic reticulum (ER)-stress pathway, and the stress-activated protein kinases/c-Jun amino-terminal kinases (SAPK/JNK) pathway. Furthermore, we have searched for substances that protect neurons from Zn^2+^-induced neurotoxicity among various agricultural products and determined carnosine (β-alanyl histidine) as a possible therapeutic agent for vascular dementia.

## 1. Introduction

Senile dementia is a serious problem for our rapidly aging society. It is characterized by profound memory loss and inability to form new memories in older adults, and its prevalence increases with age. Senile dementia is mainly divided into Alzheimer’s disease (AD), vascular dementia (VD), and dementia with Lewy bodies (DLB). Both AD and DLB are characterized by the deposition of abnormally accumulated proteins; β-amyloid protein (AβP) in AD and α-synuclein in DLB [1,2]. However, VD is a degenerative cerebrovascular disease, which is caused by a series of strokes or ischemia [3]. Worldwide, about 50 million people are affected by dementia at 2019, and nearly 10 million cases are annually increasing [4].

Increasing evidence suggests that metal dyshomeostasis is involved in the pathogenesis of AD, VD, and DLB [5,6,7]. It is widely accepted that zinc (Zn) plays critical roles in neurodegeneration after ischemia and in the pathogenesis of VD [8]. Here, we focus on the link between Zn and the pathogenesis of VD and review the molecular mechanism of Zn^2+^-induced neurotoxicity based on our and numerous other studies. We have already demonstrated the implications of the energy production pathway, the disruption of calcium homeostasis, and the endoplasmic reticulum (ER)-stress pathway in the molecular mechanism of Zn^2+^-induced neurotoxicity [9]. Based on our recent findings about the involvements of the production of reactive oxygen species (ROS) and the stress-activated protein kinases/c-Jun amino-terminal kinases (SAPK/JNK) pathway [10], we developed a hypothetical scheme about these molecular pathways of Zn^2+^-induced neurotoxicity.

Furthermore, substances that attenuate Zn^2+^-induced neurotoxicity may become potential drugs for the treatment or prevention of VD [11]. Based on this idea, we have developed a convenient screening system for such substances, and examined extracts of various agricultural products, e.g., fruits, vegetables, and fish. Among the tested substances, we found that carnosine (β-alanyl histidine) protected neurons from Zn^2+^-induced neurotoxicity [12]. Carnosine is an endogenous dipeptide, which possesses various advantageous properties such as antioxidant, antiglycation, and anticrosslinking [13,14]. Carnosine is highly accumulated in skeletal muscles and olfactory bulbs in the brain. Since the olfactory bulb is a gateway for external information and substances, carnosine is considered to be an endogenous neuroprotective substance, namely a ‘neuroprotector’ [15]. The supplement therapy of carnosine or its analogues are reportedly effective for the treatments of diabetes [16], cataract [17], and brain-related disorders such as depression and Gulf-war illness [18,19]. We have developed a system for quantitative analysis of carnosine using high-performance liquid chromatography (HPLC) [20] and demonstrated here the developmental changes of the amount of carnosine and its analogues in the brain. We also discussed the perspectives of carnosine supplement therapy for the prevention of VD.

## 2. Zinc and Vascular Dementia

VD is regarded as the second most common type of senile dementia [21]. Its risk factors are age, male sex, diabetes, and high blood pressure. VD is linked with neurodegeneration that occurs after stroke or ischemia [3,21]. After transient global ischemia, the interruption of blood flow and the resulting deprivation of oxygen and glucose induce abnormal neuronal excitation in most parts of brain, followed by excessive release of glutamate into the synaptic clefts. The successive entry of large quantities of Ca^2+^ triggers delayed death of vulnerable neurons in the hippocampus or cerebral cortex and causes the development of an infarct, which finally leads to cognitive dysfunction and VD pathogenesis. An epidemiological study has reported the exhibition of dementia symptoms in about 30% of stroke patients after 3 years [22].

Increasing evidence suggests the involvement of Zn in neuronal death after ischemia [7,23,24,25]. Zn is the second most abundant trace element in the brain. High concentrations of Zn accumulate especially in the hippocampus, cerebral cortex and amygdala [26]. Zn acts as a cofactor in more than 300 enzymatic functions and plays essential roles in normal brain functions. Moreover, a considerable amount of Zn forms free Zn ions (Zn^2+^) that are stored in presynaptic vesicles of glutamatergic neurons. In addition to glutamate, synaptic Zn^2+^ is secreted from the vesicles to the synaptic clefts during neuronal excitation [27,28]. Then, it binds to several receptors including *N*-methyl-D-aspartate (NMDA)-type glutamate receptors, γ -aminobutyric acid (GABA) receptors, and glycine receptors to regulate the neuronal excitability. Zn^2+^ also binds to various channels including Ca^2+^ channels, K^+^ channels and modulate their activities [29]. Although the precise role of synaptic Zn^2+^ is still under investigation, it is widely accepted that Zn^2+^ plays significant roles in information processing and memory formation [30,31]. Takeda et al. demonstrated that synaptic Zn2+ is involved in the maintenance of memory formation and the AβP neurotoxicity [32]. Indeed, Zn^2+^ has been reported to influence synaptic plasticity via inhibition of long-term potentiation (LTP). The disorder of ZnT3, the Zn transporter, which involves in the accumulation of Zn at synaptic vesicles, is related to the pathogenesis of AD and other neurodegenerative diseases [33]. Zn deficiency causes retardation in mental and physical development and learning disorders in infants [34]. Low synaptic Zn^2+^ reportedly enhanced the susceptibility to febrile seizure [35]. Moreover, some neurons have been reported to contain Cu^2+^ in synaptic vesicles in addition to Zn^2+^ [36]. Synaptic Cu^2+^ is also released to the synaptic clefts during neuronal excitation and bind to NMDA-type glutamate receptors, to regulate synaptic functions as well as Zn^2+^. Therefore, crosstalk between Zn^2+^ and Cu^2+^ plays important roles in physiological and pathophysiological functions in the brain [37].

Despite its importance, excess Zn^2+^ is toxic. In physiological condition, the concentration of free Zn^2+^ is strictly regulated [29]. In ischemic conditions, a considerable amount of Zn^2+^ (up to 300 µM) is reportedly co-released with glutamate into synaptic clefts following membrane depolarization [38]. Koh et al. have demonstrated that Zn accumulates in apoptotic neurons in the hippocampus after ischemia [39]. Administration of calcium ethylenediaminetetraacetic acid (Ca-EDTA), a membrane-impermeable Zn^2+^ chelator, protected hippocampal neurons after transient global ischemia in experimental animals and reduced the infarct volume [40]. Kitamura et al. have revealed an increase in extracellular Zn^2+^ in transient middle cerebral artery occlusion model (MCAO) rats using a microdialysis method [41]. Furthermore, Zn^2+^ contributes to the increased permeability of the blood brain barrier after ischemia [42].

The entry of Zn^2+^ and the increase in intracellular Zn^2+^ levels ([Zn^2+^]_i_), namely “Zn translocation”, may be the primary event in Zn^2+^-induced neurotoxicity. There are three major routes of Zn^2+^ entry: voltage-gated Ca^2+^ channels (VGCCs), NMDA-type glutamate receptors (NMDA-Rs), and AMPA/kainate-type glutamate receptors (A/K-Rs) [8]. Under normal conditions, most hippocampal neurons express AMPA receptors containing the GluR2 subunit, which are poorly permeable to Ca^2+^ and Zn^2+^. However, after ischemia, an acute reduction in GluR2 expression occurs and the neurons express a specific type of Ca^2+^-permeable AMPA receptors (Ca-A/K-R). As the permeabilities of Zn^2+^ and Ca^2+^ through Ca-A/K-R channels are greater than through NMDA receptor channels, increased expression of Ca-A/K-R channels enhances the toxicity of Ca^2+^ and Zn^2+^. Zn^2+^ is also implicated in the transcriptional regulation of Ca-A/K-R channels, as Ca-EDTA attenuated the ischemia-induced downregulation of the *GluR2* gene [40].

The Zn and Cu levels in the synaptic clefts are estimated to be 1–100 μM and 15 μM, respectively [43,44]. The concentration of Zn in the cerebrospinal fluid (CSF) has been reported to be increased by about 10–20 ppb (0.15–0.31 µM) after ischemia [41]. Considering that the synaptic cleft is a small compartment with 120-nm radius and 20-nm height and that the total volume is estimated to be 1% of the extracellular space [45], the concentration in the synaptic clefts may be much higher compared with that in the CSF.

## 3. Zn^2+^-Induced Neurotoxicity

### 3.1. GT1–7 Cells as a Model System for Investigating Zn^2+^-Induced Neurotoxicity

The molecular mechanism of Zn^2+^-induced neuronal death is of great importance for the development of drugs for VD. To this end, primary cultured neurons of rat cerebral cortex or hippocampus, or PC-12 cells, a pheochromocytoma cell line, have been used by many researchers [46,47]. However, the roles of Zn^2+^ are highly complex in these neuronal cells, which possess glutamate receptors. Since glutamate also causes neurotoxicity and Zn^2+^ regulates the glutamate-induced excitability, it is difficult to distinguish between the effects of Zn^2+^ and glutamate. We found that Zn^2+^ induced marked death of GT1–7 cells (immortalized hypothalamic neurons), as shown in Figure 1A, and that these cells were highly sensitive to Zn^2+^ compared with other neuronal cells, including primary cultures of the rat cerebral cortex or hippocampus neurons, PC-12 cells, and B-50 cells [48,49]. GT1–7 cells were originally developed by Mellon et al. in 1991 by genetically targeting tumorigenesis in mouse hypothalamic neurons [50]. These cells possess neuronal characteristics such as neurite extension, secretion of gonadotropin-releasing hormone (GnRH), and expression of neuron-specific proteins and receptors including microtubule-associated protein 2 (MAP2), tau protein, neurofilament, synaptophysin, GABA_A_ receptors, dopamine receptors, and L-type Ca^2+^ channels. However, GT1–7 cells do not exhibit glutamate-induced toxicity, as shown in Figure 1B, because they lack or possess low levels of ionotropic glutamate receptors [51]. Owing to these properties, we considered the GT1–7 cell line an excellent model system for investigating Zn^2+^-induced neurotoxicity.

### 3.2. Molecular Pathways Underlying Zn^2+^-Induced Neurotoxicity

We investigated the molecular mechanism underlying Zn^2+^-induced neurotoxicity using GT1–7 cells. Zn^2+^ induced apoptotic death of GT1–7 cells, which were terminal deoxynucleotidyl transferase-mediated biotinylated UTP nick end labeling (TUNEL)-positive and exhibited DNA fragmentation [48,49]. The LD_50_ was estimated to be 35 µM.

First, we tested the effects of treatment with various pharmacological agents prior to Zn^2+^ treatment of GT1–7 cells and found that neither antagonists nor agonists of excitatory neurotransmitters (D-APV, glutamate, and CNQX), nor those of inhibitory neurotransmitters (bicuculline, muscimol, baclofen, and GABA) influenced the viability of GT1–7 cells. However, several compounds including energy substrates (pyruvate and citrate), metal chelators (*o*-phenanthroline and deferoxamine), peptides, and amino acids (carnosine, anserine, and histidine) attenuated the Zn^2+^-induced death of GT1–7 cells [53,54,55,56,57] (Figure 1). Furthermore, we investigated the viability of GT1–7 cells with or without other metal ions after exposure to Zn^2+^. We found that co-exposure to sublethal concentrations of Cu^2+^ and Ni^2+^ remarkably exacerbated Zn^2+^-induced death, whereas co-exposure to Al^3+^, Gd^3+^, or Ca^2+^ attenuated the Zn^2+^-induced death of GT1–7 cells [58,59,60].

Next, we analyzed the Zn^2+^-induced genetic changes using DNA microarray analysis and real-time PCR (RT-PCR). After 4 h of exposure to Zn^2+^, various genes including metal-related genes (Zn transporter 1 (*ZnT-1*), metallothionein 1 (*MT1*), and metallothionein 2 (*MT2*)), endoplasmic reticulum (ER) stress-related genes, signal transduction-related genes, and Ca^2+^ signaling-related genes were upregulated [56,57]. Moreover, the gene expression after co-exposure to Cu^2+^ and Zn^2+^ (Cu^2+^ + Zn^2+^) was investigated [59]. We then used the substances that attenuated Zn^2+^-induced neurotoxicity as model compounds and investigated the molecular pathways of Zn^2+^-induced apoptotic neuronal death. In the following subsections we described the five pathways involved in Zn^2+^-induced neuronal death.

#### 3.2.1. Energy Production Pathway

Administration of sodium pyruvate, an energy substrate, significantly inhibited the Zn^2+^-induced death of GT1–7 cells [53]. This result is consistent with findings of other studies using primary cortical neurons [61,62], oligodendrocyte progenitor cells [63], or retinal cells [64]. Shelline and his colleagues have reported that Zn^2+^ exposure decreased the levels of NAD^+^ and ATP in cultured cortical neurons, and that treatment with pyruvate restored the NAD^+^ level [61]. Furthermore, administration of pyruvate to experimental animals reportedly attenuated the neuronal death after ischemia in vivo [65]. We demonstrated that pyruvate and citrate attenuated Cu^2+^-enhanced Zn^2+^-induced neurotoxicity of GT1–7 cells [66]. Increasing evidence suggests that Zn^2+^ is localized within mitochondria and triggers impaired mitochondrial functions [67,68]. Therefore, energy failure and inhibition of glycolysis in mitochondria may be involved in Zn^2+^-induced neurotoxicity.

#### 3.2.2. Ca^2+^ Homeostasis

Our pharmacological approach demonstrated that nimodipine, an L-type VGCC blocker, attenuated Zn^2+^-induced neurotoxicity. Consistently, Kim et al. have reported that Zn^2+^-induced death in PC-12 cells was attenuated by nimodipine, and enhanced by the L-type VGCC activator, S(−)-Bay K 8644 [69]. Other L-type Ca^2+^ channel blocker, nifedipine, also attenuated Zn^2+^-induced neurotoxicity [70]. Additionally, Kim et al. demonstrated that aspirin inhibited N-type VGCC and attenuated Zn^2+^-induced neurotoxicity by preventing Zn^2+^ entry [71]. These studies suggest that Ca^2+^ dyshomeostasis is involved in the mechanism of Zn^2+^-induced neurotoxicity.

To further address this issue, we used a high-resolution multisite video imaging system with Fura-2 as the cytosolic Ca^2+^ fluorescent probe to observe temporal changes in the intracellular calcium level ([Ca^2+^]_i_) after exposure to Zn^2+^ [49]. This multisite fluorometry system enables simultaneous long-term observation of temporal changes in [Ca^2+^]_i_ in more than 50 neurons. After the exposure to Zn, increases in [Ca^2+^]_i_ were observed in GT1–7 cells. As noted above, addition of Al^3+^ significantly inhibited Zn^2+^-induced neurotoxicity in a dose-dependent manner. We found that pretreatment with Al^3+^ significantly blocked the Zn^2+^-induced [Ca^2+^]_i_ elevations, though it did not influence Zn^2+^ influx. Although Al is neurotoxic, it did not exhibit toxicity in this experimental condition since Al^3+^ is difficult to enter the cells without membrane permeable chelators [72]. Thus, it is possible that Al^3+^, a known blocker of various types of Ca^2+^ channels [73], attenuates Zn^2+^-induced neurotoxicity by blocking Zn^2+^-induced elevations in [Ca^2+^]_i_.

#### 3.2.3. Endoplasmic Reticulum (ER) Stress Pathway 

Our DNA microarray analysis demonstrated that Zn^2+^ induced a marked upregulation of ER stress-related genes, including CCAAT-enhancer-binding protein homologous protein (*CHOP)*, and growth-arrest and DNA-damage-inducible gene 34 (*GADD34*) [56,57].

ER stress is associated with the accumulation of unfolded or misfolded proteins and is involved in various neurological disorders, such as cerebral ischemia, AD, and prion diseases [74]. ER stress is distinguished by three signaling proteins (ER stress sensors) termed inositol-requiring enzyme-1α (IRE1α), protein kinase R (PKR)-like ER kinase (PERK) and activating transcription factor 6 (ATF6) [75]. Upon activation, IRE1α, PERK, and ATF6 induce various signal transduction events. The phosphorylation of the α subunit of eukaryotic translation initiation factor 2α (elF2α) is mediated by PERK, and then regulates the translation of activating transcription factor 4 (ATF4). ATF4 is a transcription factor that drives the expression of *CHOP* and *GADD34*.

For more detailed analysis, we used RT-PCR to examine the Zn^2+^-induced expression of these genes and other ER stress-related genes such as immunoglobulin binding protein (*Bip*), ER degradation-enhancing α-mannosidase-like protein (*EDEM*), spliced X-box binding protein-1 (*sXBP1*), glucose-regulated protein 94 (*GRP94*), and protein disulfide isomerase (*PDI*) [59]. After 4 h of exposure to Zn^2+^ alone, enhanced expression levels of activity-regulated cytoskeleton (*Arc*), *CHOP*, *GADD34*, and *ATF4* were observed as well as metal-related genes, including *ZnT-1*, *MT1*, and *MT2*. By contrast, other ER stress-related genes including *Bip*, *EDEM*, *sXBP1*, *GRP94*, and *PDI* did not exhibit significant changes. Based on our results, it is plausible that the PERK-related pathway is involved in Zn^2+^-induced ER stress. Furthermore, a synergistic increase in the gene expression levels of *Arc*, *CHOP*, and *GADD34* was observed in cells co-exposed to Cu^2+^ and Zn^2+^, despite the fact that Cu^2+^ alone did not induce significant changes in these genes. CHOP is responsible for initiating an apoptotic cascade [76] and mediates the activation of GADD34, which reportedly increases after traumatic brain injury [77]. We also demonstrated that the amount of CHOP protein was significantly increased after Cu^2+^+Zn^2+^ treatment, compared with Zn^2+^ alone using western blot analysis [59]. Furthermore, we found that dantrolene, an inhibitor of ER stress, attenuated Zn^2+^-induced neurotoxicity [59]. Based on these findings, it is highly likely that the ER stress pathway is critically involved in Zn^2+^-induced neurotoxicity, and that low concentrations of Cu^2+^ promote Zn^2+^-induced neurotoxicity by potentiating the ER stress pathway. Moreover, we have demonstrated that ER stress is involved in the Ni^2+^-enhancement of Zn^2+^-induced neurotoxicity [60].

#### 3.2.4. SAPK/JNK Pathway

Our DNA microarray results demonstrated that several genes downstream of the stress-activated protein kinases/c-Jun amino-terminal kinases (SAPK/JNK) pathway, such as *c-Jun* and *ATF2*, were upregulated in cells exposed to Cu^2+^ + Zn^2+^. SAPK/JNK are members of the mitogen-activated protein kinase (MAPK) family. The SAPK/JNK signaling pathway plays critical roles in apoptotic cell death, necroptosis, and autophagy [78]. This pathway has been shown to be activated by a variety of environmental stressors, such as oxidative stress, inflammatory cytokines, and metals. Upon activation of this pathway, MAPK kinase 4 (MKK4) or MKK7 phosphorylates and activates the SAPK/JNK. Thereafter, c-Jun and ATF2, major downstream factors of SAPK/JNK, are phosphorylated and activated by SAPK/JNK. Ultimately, the phosphorylated forms of c-Jun and ATF2 induce downstream factors related to cell death and mitochondrial injury, leading to cell death. We examined the expression of these factors and we found that phosphorylated (i.e., active) forms of SAPK/JNK were increased by Cu^2+^ + Zn^2+^ treatment in GT1–7 cells [10]. Consistently, phospho-c-Jun and phospho-ATF2 were also induced by Cu^2+^ + Zn^2+^ co-treatment. Moreover, SP600125, an inhibitor of the SAPK/JNK signaling pathway, significantly suppressed the activation of the SAPK/JNK signaling pathway by Cu^2+^ + Zn^2+^ and the neuronal cell death.

#### 3.2.5. ROS Pathway

Oxidative stress is involved in various degenerative pathways; reactive oxygen species (ROS) induce the SAPK/JNK pathway, the ER stress pathway, and numerous other adverse effects [79,80]. Zn exists only as Zn^2+^ and is not implicated in the redox pathway, although it is reportedly linked with oxidative stress. In contrast, Cu is a redox-active metal that exists as oxidized Cu^2+^ and reduced Cu^+^. We found that the antioxidant, thioredoxin-conjugated human serum albumin (HSA-Trx), attenuated neuronal death induced by Cu^2+^ + Zn^2+^ [81]. We also found that the addition of Cu^2+^ induced ROS production in GT1–7 cells, despite that Zn^2+^ alone did not produce ROS nor did it influence Cu^2+^-induced ROS production [10]. Thus, it is possible that Cu^2+^ triggers ROS production, and then induces the SAPK pathway and/or the ER pathway, thereby enhancing Zn^2+^-induced neurotoxicity.

### 3.3. Hypothesis Regarding the Molecular Pathways Underlying Zn^2+^-Induced Neurotoxicity

Based on these results, we propose a possible scheme of the pathways underlying Zn^2+^-induced neurotoxicity (Figure 2).

In the case of transient global ischemia, neuronal excitation occurs in a large area of the brain. Thereafter, both Zn^2+^ and Cu^2+^ are secreted from synaptic vesicles into the synaptic clefts and translocate into the neurons. Chelators such as Ca-EDTA and *o*-phenanthroline block this translocation process and attenuate Zn^2+^-induced neuronal death. Furthermore, the increased [Zn^2+^]_i_ triggers the inhibition of the energy production machinery in the mitochondria. The energy substrates, pyruvate and citrate, prevent this process. Zn^2+^ also leads to an increase in [Ca^2+^]_i_ levels, which is inhibited by Al^3+^ and other Ca^2+^ channel blockers. The increase in [Ca^2+^]_i_ induces ROS production and triggers the ER stress and various apoptotic pathways including the SAPK/JNK pathway. Co-exposure to Cu^2+^ further induces ROS production and enhances the ER stress pathway and/or the SAPK/JNK pathway. Finally, these processes trigger neurodegenerative pathways, and lead to the neuronal death observed in VD. Carnosine is released from glial cells and enters into neurons by peptide transporters. Carnosine regulates Zn^2+^ and Cu^2+^ homeostasis, inhibits Zn^2+^-induced ER stress and ROS production, and protects neurons from excess Zn^2+^ (details are discussed in Section 4.4). 

## 4. Carnosine as a Protective Substance Against Zn^2+^-Induced Neurotoxicity

### 4.1. Screening System for Protective Substances Against Zn^2+^-Induced Neurotoxicity

Considering the involvement of Zn in ischemia-induced neuronal death, substances that protect against Zn^2+^-induced neuronal death may be potential candidates for the prevention or treatment of neurodegeneration following ischemia, and ultimately provide a lead for VD treatments. To explore this idea, we established a convenient and rapid screening system for such substances using GT1–7 cells [10]. We examined the potential inhibitory effects of various agricultural products such as vegetable extracts, fruits extracts, and fish extracts. Among tested, we found that extracts of Japanese eel (*Anguilla japonica*), mango fruit (*Mangifera indica* L.), and round herring (*Etrumeus teres*) protected GT1–7 cells from Zn^2+^-induced neurotoxicity, then separated the active fraction using HPLC and determined the components’ structures by LC-mass spectrometry (MS). Finally, the active compounds were revealed to be carnosine, citrate, and histidine [12,56,82].

### 4.2. Carnosine as an Endogenous Neuroprotector

Carnosine is a dipeptide composed of β-alanine and histidine. Figure 3 shows the chemical structures of carnosine and its analogues (anserine (1-methyl carnosine) and homocarnosine). They are naturally occurring dipeptides commonly present in most vertebrate tissues such as birds, fishes, and mammals including humans [13,14]. Carnosine is found in particularly high concentrations in animals and fish that exhibit high levels of exercise, such as horses, chickens, bonitos, and whales. Carnosine is one of the most abundant small-molecule compounds in skeletal muscles with concentrations ranging from 50 to 200 mM, similar to those of creatine and ATP [83]. During high-intensity anaerobic exercise, muscle contractions lead to lactic acid production and a decrease in intracellular pH, which influences various metabolic functions, and the resulting acidosis caused muscle contractile fatigue. Owing to the alkaline property of carnosine (p*K*a value was 7.01), it is thought to play a significant role in intracellular buffering and in maintaining the pH balance in the muscle [84]. Therefore, the concentration of muscle carnosine is considered to have a positive relationship with exercise performance. Indeed, highly trained athletes have higher carnosine levels than those in untrained individuals [85]. Compared with humans, horses have a 6–10 times higher muscle carnosine concentration. We analyzed the amount of carnosine in five muscles of thoroughbred horses and found that the amount of carnosine is linked to the muscle fiber type [86]. Furthermore, dietary supplementation of carnosine or β-alanine induces an increase in the concentration of muscle carnosine and a delay in fatigue during high-intensity exercise [87]. 

Additionally, carnosine possesses various beneficial characteristics such as antioxidation, antiglycation, anticrosslinking, and metal chelation [14]. Carnosine scavenges both reactive oxygen and nitrogen, which contain unpaired electrons. Carnosine inhibits lipid oxidation through a combination of free radical scavenging and metal chelation. It inhibits the Maillard reaction that involves reducing sugar and proteins, providing a multitude of end products, most notably advanced glycation end-products (AGEs), which contribute to the pathogenesis of various senile diseases such as AD, vascular stiffening, atherosclerosis, osteoarthritis, inflammatory arthritis, and cataracts. Carnosine and its analogues are reportedly effective for oxidative stress-induced disorders [88]. Furthermore, carnosine has anticrosslinking properties that inhibit the oligomerization of proteins. It is widely accepted that crosslinking of disease-related proteins (e.g., AβP, prion protein, and α-synuclein) and the subsequent conformational changes are central in the pathogenesis of various neurodegenerative diseases termed “conformational diseases” including AD, DLB, and prion diseases [89]. Corona et al. have demonstrated that carnosine supplementation inhibited AβP deposition and improved learning abilities of AD model mice [90]. Carnosine prevents the oxidative stress and inflammation induced by AβP [91]. We showed that carnosine attenuated the neuronal death induced by prion protein fragment peptide (PrP106–126) by changing its conformation [92]. Since carnosine inhibits the fibrillization of lens α-crystalline, N-acetyl carnosine is used as a drug for cataract treatment [17]. Carnosine is a chelator of metal ions and forms complexes with Ca^2+^, Cu^2+^, and Zn^2+^. Thus, it is likely that this dipeptide controls the availability of Zn^2+^ in neuronal tissue, particularly in the olfactory lobe where both carnosine and Zn are enriched. The Zn-carnosine complex, termed polaprezinc, is effective in the repair of ulcers and other lesions in the alimentary tract [93]. Polaprezinc is also used for Zn supplement therapy and is protective against the cadmium-induced lung injury [94]. Additionally, carnosine reportedly attenuates Mn-induced neurotoxicity [95]. Based on these beneficial characteristics, carnosine is considered to be a “gatekeeper” or “neuroprotector” in the brain [15].

### 4.3. Carnosine in the Brain

The concentrations of carnosine and related peptides are different among species and vary according to regions [14]. To explore the significance of carnosine and related compounds, we have established a convenient system for quantitative analysis of carnosine, anserine, and homocarnosine using HPLC [20]. Since carnosine and its analogues are highly hydrophilic, it is difficult to separate these compounds using reversed-phase HPLC equipped with a conventional octadecylsilyl (ODS) column, which is generally used in peptide analysis. Thus, we used an HPLC system equipped with a carbon column (Hypercarb^TM^), which contains porous graphite carbon. Using this system, we have analyzed the amount of carnosine and its analogues in various foods using conventional UV spectroscopy at 215 nm [9].

We investigated their presence in the rat brain. Figure 4A shows a typical chromatogram of standard carnosine, anserine, alanyl histidine, and homocarnosine. Figure 4B exhibits a typical chromatogram of a water extract of rat olfactory bulbs dissected after 15 weeks of age, which was heated at 95 °C for 15 min to remove proteins. After this simple pre-treatment, the recovery rates of carnosine and other compounds were determined to be more than 99%. We found that the considerable amount of carnosine in olfactory bulbs. However, only small amounts of carnosine were observed in the cerebral cortex and cerebellum, and anserine was not detected in any of the tested regions. When we examined the changes in carnosine and its analogues in embryos, and rats at 6 and 15 weeks of age, we found that the amount of carnosine in olfactory bulbs increased with age (Figure 4C). Meanwhile, homocarnosine did not exhibit changes with age after birth (Figure 4D). These results are consistent with previous studies. In the mammalian brain, carnosine and homocarnosine, but not anserine, have been detected [96]. It is secreted from oligodendrocytes by the stimulation of glutamate [97]. Boldyrev et al. reviewed that carnosine is mainly present in neurons of olfactory bulbs or in glial cells, and its concentration in the olfactory bulb is more than 1000 µmol/kg [14], similar to our findings in Figure 4C. 

### 4.4. The roles of Carnosine in Protection from Zn^2+^-Induced Neurotoxicity

We found that carnosine protected neurons against Zn^2+^-induced neurotoxicity, and then investigated how it affects the pathways. As carnosine can chelate Zn^2+^, it is plausible that it binds extracellular Zn^2+^ and inhibits Zn^2+^ translocation like other chelators. However, our results using a Zn^2+^-sensitive fluorescent dye and RT-PCR indicated that carnosine did not influence [Zn^2+^]_i_ nor the expression of ZnT-1 [57]. Meanwhile, we demonstrated that carnosine inhibited Zn^2+^-induced upregulation of ER stress-related genes such as *GADD34* and *CHOP*, and attenuated neurodegeneration induced by ER-stressors such as thapsigargin and tunicamycin [57]. Thus, it is possible that carnosine protects neurons from Zn^2+^ by affecting the ER stress pathway, not by inhibiting Zn^2+^ translocation. Based on the activities of carnosine, we have published two patents for carnosine and related compounds (D-histidine) as drugs or supplements for the prevention and/or treatment of VD [98,99]. Moreover, increasing evidence from experimental animals suggests that carnosine protects against ischemia-induced neurodegeneration in vivo [100,101,102,103].

It is widely believed that orally administered carnosine rapidly degraded to β-alanine and histidine by carnosinase in the blood. However, the supplementation of carnosine or β-alanine reportedly increased the level of carnosine in the brain [104,105]. Therefore, it is possible that dietary carnosine or related amino acids can be synthesized to carnosine in the brain and enters into cells by oligopeptide transporters such as PEPT2, PHT1, and PHT2 [106]. Considering the carnosine level in the body decreases with age [107], the carnosine supplementation therapy may be beneficial for VD, AD, and other diseases. The supplementation of carnosine (40 mg/day) was effective in treatment of patients with major depressive disorders [18]. Recent epidemiological study revealed the inverse correlation between serum β-alanine and pathogenesis of dementia [108]. Furthermore, the carnosine/anserine supplementation (750 mg anserine and 250 mg carnosine per day) reportedly improved episodic memory in elderly people [109] or mild cognitive impairment [110].

## 5. Conclusions and Future Perspectives

Our hypothesis about the molecular pathways of Zn^2+^-induced neurotoxicity may aid for the developments of preventive drugs for VD. Carnosine has many beneficial properties such as water solubility, heat-inactivation, and being nontoxic, and therefore, it may become a good neuroprotective drug or supplement that is beneficial for our health. Further research on the molecular mechanism by which carnosine prevents neurotoxicity is required.

## Figures and Tables

**Figure 1 ijms-21-02570-f001:**
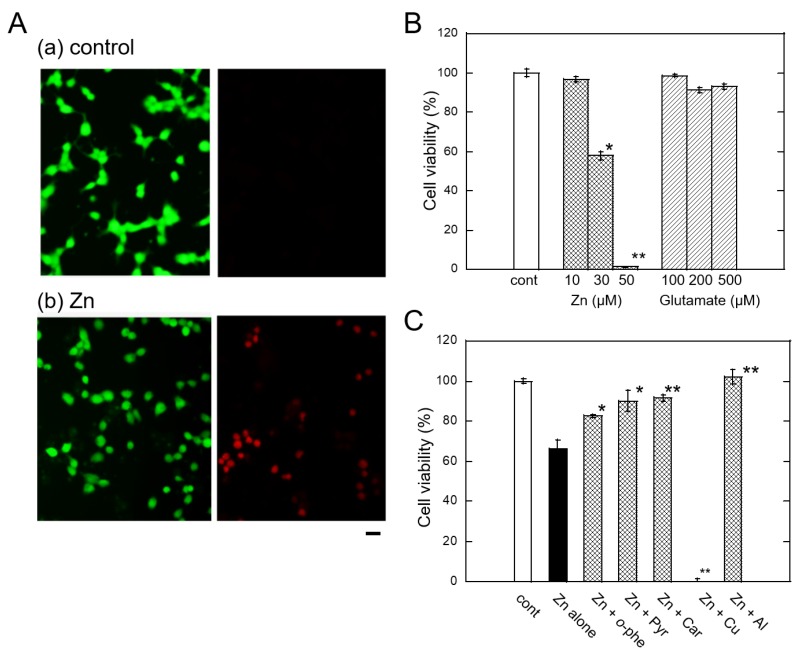
Zn^2+^-induced neurotoxicity in GT1–7 cells. (**A**) GT1–7 cells were exposed to with or without 30 µM ZnCl_2_. After 24 h, live cells were stained with carboxy fluorescein (green) and dead cells were stained with propidium iodide (red). Bar represents 50 µM. (**B**) Viability of GT1–7 cells after exposure to Zn or glutamate. GT1–7 cells were treated with ZnCl_2_, or glutamate. After 24 h, cell viability was analyzed using the WST-1 method, which counts the living cell number by measures the cellular mitochondrial dehydrogenase activity [52]. Data are expressed as mean ± SEM, *n* = 6. * *p* < 0.05, ** *p* < 0.01, compared with control. (**C**) Viability of GT1–7 cells after exposure to Zn with other pharmacological substances. GT1–7 cells were treated with 30 µM Zn with sodium pyruvate (Pyr: 1 mM), *o*-phenanthroline (*o*-phe; 20 µM) carnosine (Car: 1 mM), CuCl_2_ (10 µM), or AlCl_3_ (100 µM). After 24 h, cell viability was analyzed using the WST-1 method. Data are expressed as mean ± SEM, *n* = 6. * *p* < 0.05, ** *p* < 0.01 compared with Zn alone.

**Figure 2 ijms-21-02570-f002:**
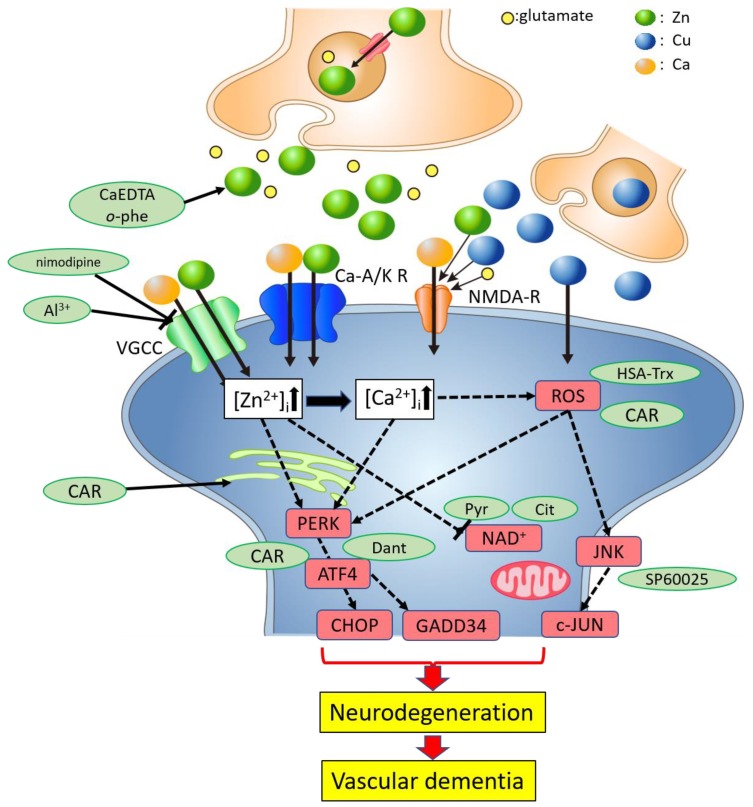
Schematic of the hypothetical molecular mechanisms underlying the protective effects of carnosine against neuronal death induced by Zn. Zn and Cu are stored in presynaptic vesicles and secreted to the synaptic clefts during ischemia. Excess amounts of secreted Zn can translocate into cells and induce disruption of Ca^2+^ homeostasis, energy failure in mitochondria, ER stress and oxidative stress, and apoptotic neuronal death. Co-exposure to Cu^2+^ enhances these effects. These pathways are inhibited by Zn chelators (Ca-EDTA and *o*-phenanthroline (*o*-phe)), Ca^2+^ channel blockers (Al^3+^ and nimodipine), energy substrates (pyruvate (Pyr) and citrate (Cit)), and the inhibitor of the ER stress pathway (dantrolene (Dant)) or the inhibitor the SAPK/JNK signaling pathway (SP600125), antioxidants (HSA-Trx), or carnosine (CAR). Carnosine is synthesized in glial cells and is secreted in response to stimulation by glutamate and Zn. It protects neurons from Zn neurotoxicity. Carnosine inhibits the ER stress-related apoptotic pathways and the ROS pathway.

**Figure 3 ijms-21-02570-f003:**
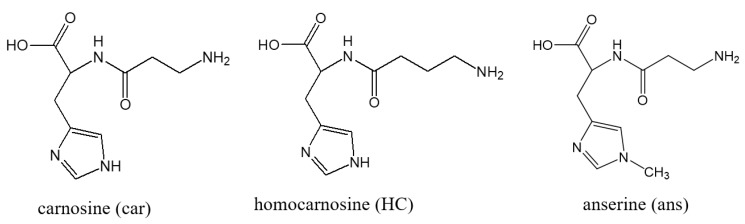
Structures of carnosine and its analogues. Chemical structures of carnosine (Car), anserine (Ans), and homocarnosine (HC) are exhibited.

**Figure 4 ijms-21-02570-f004:**
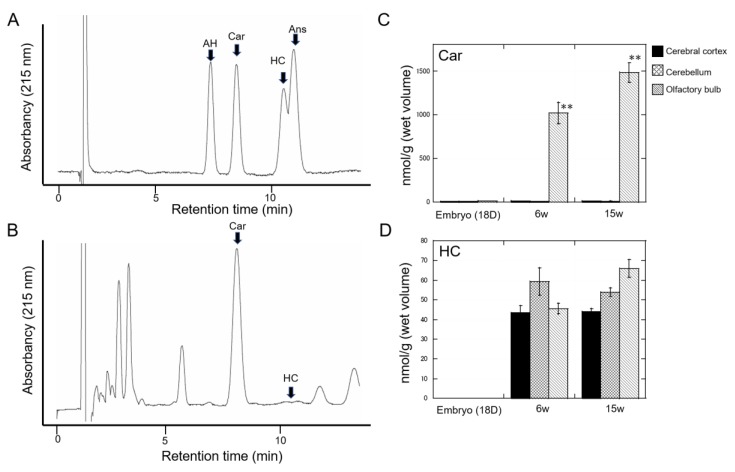
Developmental changes in carnosine and related compounds in the brain. Each brain region (cerebrum, cerebellum, and olfactory bulb) was dissected from Wistar rats, homogenized and heated at 95 °C for 15 min. After centrifugation, the supernatant was analyzed by HPLC equipped with a Hypercarb^TM^ column (Thermo Fisher Science, Waltham, MA, USA). Carnosine (Car), homocarnosine (HC), alanyl histidine (AH), and anserine (Ans) were analyzed using an isocratic elution of 7% acetonitrile in the presence of 0.1% trifluoroacetic acid at a flow rate of 1 mL/min and monitored with UV at 215 nm. Data are expressed as mean ± S.E.M (*n* = 4). (**A**) Typical chromatogram of standard solutions. (**B**) Typical chromatogram of the olfactory bulb extract at 15 weeks. (**C**) Developmental changes in carnosine. The amount of carnosine (nmol/g wet tissue) in cerebral cortex, cerebellum, and olfactory bulb at embryo 18 days (E18), and 6 weeks (6 W) and 15 weeks (15 W) postnatal was analyzed by HPLC. Data are expressed as mean ± S.E.M (*n* = 4). * *p*<0.05, compared with E18. (**D**) Developmental changes in homocarnosine. The amount of homocarnosine (nmol/g wet tissue) in cerebral cortex, cerebellum, and olfactory bulb at embryo 18 days (E18), and 6 weeks (6 W) and 15 weeks (15 W) postnatal was analyzed by HPLC. Data are expressed as mean ± S.E.M (*n* = 4).

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
