# Peer review of "Carnosine as a Possible Drug for Zinc-Induced Neurotoxicity and Vascular Dementia"

_ijms, 2020, doi:10.3390/ijms21072570_

Round 1

Reviewer 1 Report

The review of Masahiro Kawahara et al. is an attempt to highlight the possible neuroprotective role of carnosine in vascular dementia. The topic could be interesting, but the way it was developed is not convincing at all.

1) The Authors mostly refer to their own data with very poor analysis of the literature about this argument. This is quite unusual, especially when considering that some interesting report such as those describing carnosine neuroprotection in other neurodegenerative disease are not mentioned at all.

2) The title does not properly fit with the text, as only zinc (title: metal-targeted approach) and vascular dementia (title: …. for neurodegenerative diseases) are treated.

3) Carnosine as an endogenous neuroprotector, as well as its possible role in Zn-mediated neurotoxicity, is described in a poor manner, and again data from the Authors are often reported, while data from the literature omitted.

4) Introduction, senile dementia: only info about Japan in 2013 (seven years ago!) are described. These should be updated and compared with those worldwide (Europe, USA).

5) Line 57, again only data of Japan are stated. Also this should be extended.

6) Line 148, the use of dose is inappropriate as in cell culture concentration should be used instead.

7) Line 189, details about the mechanism according which aspirin prevent Zn entry through calcium channels would be welcome.

Author Response

Thank you very much for your comments and suggestions. I believe my paper can be improved based on your suggestions

1)The Authors mostly refer to their own data with very poor analysis of the literature about this argument. This is quite unusual, especially when considering that some interesting report such as those describing carnosine neuroprotection in other neurodegenerative disease are not mentioned at all.

Reply: Thank you very much for your suggestions. I am sorry for the incomplete literatures. I added 28 references based on your and other reviewer’s comments and discussed more about the neuroprotective roles of carnosine (line 51-55, line 336-358, line 411-423.)

2)The title does not properly fit with the text, as only zinc (title: metal-targeted approach) and vascular dementia (title: …. for neurodegenerative diseases) are treated.

Reply: Thank you very much for your suggestion. I changed the title to “Carnosine as a possible drug for zinc-induced neurotoxicity and vascular dementia”.

3)Carnosine as an endogenous neuroprotector, as well as its possible role in Zn-mediated neurotoxicity, is described in a poor manner, and again data from the Authors are often reported, while data from the literature omitted.

Reply: Thank you very much for your suggestions. I added references and changed the location of sentences, and added detailed discussion (line 51-55, line 336-358, line 411-423).

4)Introduction, senile dementia: only info about Japan in 2013 (seven years ago!) are described. These should be updated and compared with those worldwide (Europe, USA).

Thank you very much for your suggestion. I added ref 4 and discussed world wide prevalence in line 33-34.

5)Line 57, again only data of Japan are stated. Also this should be extended.

Reply: Thank you for your suggestion. I changed the explanation in line 62 and added Ref 21.

6)Line 148, the use of dose is inappropriate as in cell culture concentration should be used instead.

Reply: Thank you for your correction. I deleted and changed the explanation (line 157)

7) Line 189, details about the mechanism according which aspirin prevent Zn entry through calcium channels would be welcome.

Reply: Thank you for your comments. In Ref60, Kim et al. found that aspirin attenuated Zn-induced toxicity and aspirin inhibited VGCC. However, the detailed mechanism how aspirin inhibits VGCC was under investigation, and as far as I know, not reported afterwards. I changed line 196-197.

Reviewer 2 Report

The work is very well designed and well written. I think it is suitable for pubblication with minor revisions

Line 42, have you already demonstrated this mechanism? if yes, add the corresponding reference.

Line 53, "discuss" may be corrected in "discussed"

Line 363 to correct “to reduce to remove”

I would add, if disposable, some data about the bioavailability of carnosine.

In addition I would suggest to report the dose, if disposable, of carnosine used in the clinical trial reported in the literature. 

Author Response

Thank you very much for your kind comments and suggestions. Owing to your comments, I could improve my manuscripts.

1) Line 42, have you already demonstrated this mechanism? if yes, add the corresponding reference.

Reply; Thank you very much for your comments. I changed the reference here (Ref.10). Details are discussed in line 253-256.

2) Line 53, "discuss" may be corrected in "discussed"

 Reply: I am sorry for the mistake. I corrected as suggested in line 58.

3) Line 363 to correct “to reduce to remove”

 Reply: I am sorry for the mistake. I corrected as suggested (line372).

4) I would add, if disposable, some data about the bioavailability of carnosine.

 Reply: Thank you for your comments. The orally administered carnosine rapidly degraded and not detected in the blood. However, brain calnosine contents are increased after orally administration (Ref.103,104). So, it is possible that carnosine is re-synthesized from histidine and beta alanine in the brain. I added comments and discussed about the fate of carnosine in 413-417.

5) In addition I would suggest to report the dose, if disposable, of carnosine used in the clinical trial reported in the literature. 

Reply: Thank you for your comments. I added the dose of carnosine etc. in the therapy in line419-423 and discussed. The amount (about 1g of carnosine) seems to be reasonable, since commercially available carnosine-containing supplements are contained ~200mg of tablet.

Reviewer 3 Report

The authors describe the current data pointing at a beneficial effect of carnosine for vascular dementia. The major effect of carnosine in this context is related to the protection against Zn neurotoxicity. The review starts describing the connections between vascular dementia and abnormal Zn concentration. Then, the authors summarize their contribution to unraveling Zn-mediated neurotoxicity in this field. 

Minor comments:

-The literature describing the role of metals in neurodegeneration should be updated to more recent papers. Seminal work in the field was indeed performed in the late '90s early 2000 but the authors should try to nevertheless give a more up to date overview.  

-The sentence at lines 104-107 is not very clear to me. Do the authors mean the sum of the synaptic clefts is equal to  50% of the volume of the brain? In which sense does this correlate with the potential lower Zn levels in the CSF?

-Fig 1: the authors should describe the WST-1 method as many readers might not be familiar with that. The resolution of the images in panel A should be increased: it seems out of focus.

-The authors mention that Al can block the Zn-induced [Ca]i elevations and thereby attenuate Zn-induced neurotoxicity. I think in this context it would be worth mentioning that Al per se is also neurotoxic and there is a lot of papers describing a potential detrimental accumulation of this metal in AD. 

-In section 4.3 the authors describe the amount of carnosine in the brain referring almost exclusively to their previous papers. This part should be integrated with other studies. A reference for the statement at line 372 is missing. 

-The section "Conclusion and future perspective" is currently very synthetic. The authors should elaborate more on the points raised, especially in relation to the potential value of carnosine and related compounds for therapeutics.   

Author Response

Thank you very much for your comments and suggestions. I believe, owing to your help, I could polish up and improve my article.

1) The literature describing the role of metals in neurodegeneration should be updated to more recent papers. Seminal work in the field was indeed performed in the late '90s early 2000 but the authors should try to nevertheless give a more up to date overview.  

Reply: Thank you very much for your comments. I added several newer references 23-37 and discussed in line 79-94.

2) The sentence at lines 104-107 is not very clear to me. Do the authors mean the sum of the synaptic clefts is equal to  50% of the volume of the brain? In which sense does this correlate with the potential lower Zn levels in the CSF?

Reply: I am sorry for the mistake. I corrected to “1% of the extracellular space of the brain” (line117-118). Thus, if Zn increases 10 nM in CSF, the concentration in the synaptic clefts may be 1 uM, and may be much higher.

3) Fig 1: the authors should describe the WST-1 method as many readers might not be familiar with that. The resolution of the images in panel A should be increased: it seems out of focus.

Reply: Thank you for the comments. WST-1 method, a modified MTT assay using water-soluble hormazan, is used for the counting living cells by measuring mitochondorial dehydrogenase activity. I added the explanation in line 147-148 and the reference No.53. Since the original photo of Fig.1A is huge, I must use low resolution file to paste in word file. So, I re-organized Fig.1 and dissected higher and smaller resolution images.

4) The authors mention that Al can block the Zn-induced [Ca]i elevations and thereby attenuate Zn-induced neurotoxicity. I think in this context it would be worth mentioning that Al per se is also neurotoxic and there is a lot of papers describing a potential detrimental accumulation of this metal in AD. 

I am sorry for the lack of explanation. It is true that Al is neurotoxic. I myself has investigated its neurotoxicity. However, in this experimental condition (AlCl3, 100~250 uM for 1 day), little amount of Al can enter the cell except with membrane permeable chelator such as maltol. I have confirmed that Al itself did not cause neurotoxicity (Ref. 72). I added the comments in 206-207.

5) In section 4.3 the authors describe the amount of carnosine in the brain referring almost exclusively to their previous papers. This part should be integrated with other studies. A reference for the statement at line 372 is missing. 

Thank you for your comments. I added comments about the carnosine amount in line 380-385. Reference at line372 was moved to No.15 and discussed in 336-360, based on other reviwer’s comments.

6) The section "Conclusion and future perspective" is currently very synthetic. The authors should elaborate more on the points raised, especially in relation to the potential value of carnosine and related compounds for therapeutics.   

Reply; Thank you very much for your comments. Based on your comments and that from other reviwer’s, I moved and added explanations and future possibility about carnosine supplementary in line 417-427.

Round 2

Reviewer 1 Report

The Authors made significative improvements to the manuscript which is now suitable for publication.